# EMBEDDING COMPRESSION WITH HASHING FOR EFFICIENT REPRESENTATION LEARNING IN GRAPH

## ABSTRACT

Graph neural networks (GNNs) are deep learning models designed specifically for graph data, and they typically rely on node features as the input node representation to the first layer. When applying such type of networks on graph without node feature, one can extract simple graph-based node features (e.g., number of degrees) or learn the input node representation (i.e., embeddings) when training the network. While the latter approach, which trains node embeddings, more likely leads to better performance, the number of parameters associated with the embeddings grows linearly with the number of nodes. It is therefore impractical to train the input node embeddings together with GNNs within graphics processing unit (GPU) memory in an end-to-end fashion when dealing with industrial scale graph data. Inspired by the embedding compression methods developed for natural language processing (NLP) models, we develop a node embedding compression method where each node is compactly represented with a bit vector instead of a float-point vector. The parameters utilized in the compression method can be trained together with GNNs. We show that the proposed node embedding compression method achieves superior performance compared to the alternatives.

## 1 INTRODUCTION

Graph neural networks (GNNs) are representation learning methods for graph data. When a GNN model is applied on node classification problems, the model typically learns the node representation from input node features $\mathbf{X}$ and its graph $\mathcal{G}$ where the node features $\mathbf{X}$ are used as the input node representation to the first layer of the model and the graph $\mathcal{G}$ dictates the propagation of information (Kipf & Welling, 2016; Hamilton et al., 2017; Zhou et al., 2020). However, the input node features $\mathbf{X}$ may not always be available for certain datasets. In order to apply such type of model on graph without node features $\mathbf{X}$, we could either 1) extract simple graph based node features (e.g., number of degrees) from the graph $\mathcal{G}$ or 2) use embedding learning methods to learn the node embeddings as features $\mathbf{X}$ (Duong et al., 2019). While both approaches are valid, it has been shown that the second approach constantly outperforms the first one with a noticeable margin (Duong et al., 2019), and most recent methods learn the node embeddings jointly with the parameters of GNNs (He et al., 2017; 2020; Wang et al., 2019).

Learning node features (or embedding) $\mathbf{X}$ for graph with small number of nodes may not be much of a problem for common computer system. But, as the size of the embedding matrix $\mathbf{X}$ grows linearly with the number of nodes, scalability quickly becomes a problem, especially when attempting to apply such method on industrial grade graph data. For example, if a given graph has 1 billion nodes, we set the dimension of the learned embedding to 64, and store the embedding $\mathbf{X}$ using single-precision floating-point format, the memory cost for the embedding layer alone is 238 gigabytes, which is beyond the capability of common graphics processing unit (GPU). To solve the scalability issue, we adopt the embedding compression idea originally developed for natural language processing (NLP) models (Suzuki & Nagata, 2016; Shu & Nakayama, 2017; Svenstrup et al., 2017; Takase & Kobayashi, 2020).

Particularly, we study the `ALONE` method proposed by Takase & Kobayashi (2020) as it only requires single stage of training unlike other methods. `ALONE` represents each word using a randomly generated compositional code vector; then a decoder model, which can be trained end-to-end with

the downstream model[1], uncompresses the compositional code vector into a floating-point vector. The bit size of the compositional code vector is parametrization by $c$ and $m$ where $c$ is the cardinality of each element in the code vector and $m$ is the length of the code vector. For example, if we set $c = 4$ and $m = 6$, one valid code vector is $[2, 0, 3, 1, 0, 1]$ where the length of the vector is 6 and each element in the vector is within the set $\{0, 1, 2, 3\}$. The code vector can be converted to a bit vector of length $m \log_2 c$ by representing each element in the code vector as a binary number and concatenating the resulting binary numbers[2]. Continuing the example, the code vector $[2, 0, 3, 1, 0, 1]$ can be compactly stored as $[10\,00\,11\,01\,00\,01]$.

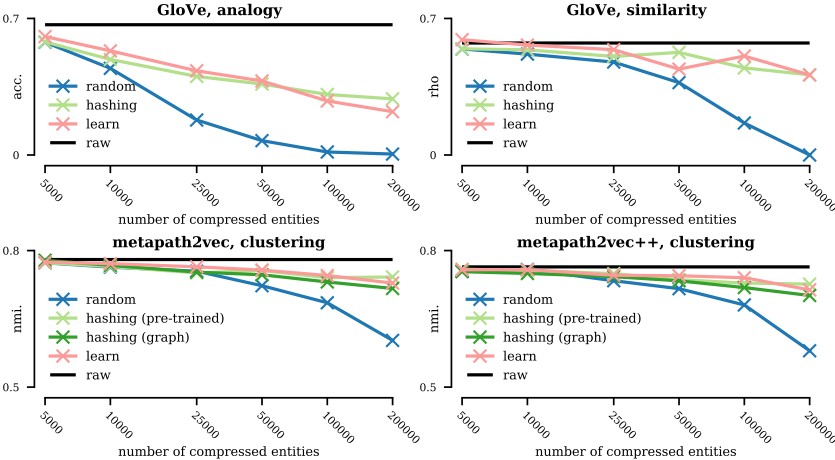

Figure 1: Three coding schemes are tested: 1) random coding/`ALONE`, 2) hashing-based coding/the proposed method, and 3) learning-based coding scheme with autoencoder. For `GloVe` embeddings, we apply the hashing-based coding method on the pre-trained embedding. For `metapath2vec` and `metapath2vec++` embeddings, we apply the hashing-based coding method on either the pre-trained embedding or the adjacency matrix from the graph. The horizontal line labeled with "raw" shows the performance of the original embeddings' performance without any compression. The y-axis of each sub-figure is the performance measurement (the higher the better). See Section 5.1 for more details.

Using the same conversion trick, it only requires 48 bits to store each word with the parametrization ($c = 64, m = 8, 8 \log_2 64 = 48$ bits) used by Takase & Kobayashi (2020) in their experiments. The coding scheme can uniquely represent up to $2^{48}$ words, which is way beyond the number of words (or sub-words) used in conventional NLP models (Vaswani et al., 2017; Takase & Okazaki, 2019). However, generating the code vectors in a random fashion hinders the quality of the uncompressed embedding. One way to quickly benchmark an embedding compression method's capability is by evaluating the performance of reconstructed (or *uncompressed*) pre-trained embeddings. As shown in Figure 1, when the model compresses more and more embeddings, the performance of the reconstructed embeddings drops considerably when each entity is represented with a randomly generated code vector following Takase & Kobayashi (2020) (see lines labeled as "random"). The phenomenon is observed in our experiments with `GloVe` word embeddings (Pennington et al., 2014b) on both word analogy/similarity task and `metapath2vec`/`metapath2vec++` node embeddings (Dong et al., 2017a) on node clustering task.

In order to solve this performance degradation problem, instead of using a randomly generated code vector to represent each entity, we adopt an efficient random projection hashing method to generate a code vector for each entity using auxiliary information such as the adjacency matrix associated with the graph $\mathcal{G}$ or the pre-trained embedding[3]. The adopted random projection hashing method is

---

[1]For the experiments conducted by Takase & Kobayashi (2020), `ALONE`'s decoder model is trained together with transformer (Vaswani et al., 2017; Takase & Okazaki, 2019) on machine translation and summarization.

[2]The conversion mechanism is more space efficient when $c$ is set to a power of 2.

[3]We only apply the hashing-based coding method on the pre-trained embedding in experiments where the pre-trained embeddings are available (i.e., Section 5.1). For other experiments, we use the adjacency matrix provided by the associated graph $\mathcal{G}$.

a locality-sensitive hashing (LSH) method (Charikar, 2002) because it hashes entities with similar auxiliary information into similar code vectors. By referring once again to Figure 1, the proposed method with hashing-based coding scheme (see lines labeled as "hashing") outperformed the random coding scheme in all scenarios. Similar to `ALONE`, the proposed method does not introduce additional training stage since it is based on random projection. On top of that, the memory footprint is identical to `ALONE` as the proposed method only replaces the coding scheme.

In additional to the proxy tasks of pre-trained embedding reconstruction, we also compare the effectiveness of different coding schemes where the GNN model and the decoder model are trained together in an end-to-end fashion. Particularly, we trained the `GraphSAGE` model (Hamilton et al., 2017) on three different node classification datasets. We choose to use the `GraphSAGE` model (Hamilton et al., 2017) because it is one of the most scalable GNNs in the literature (Ying et al., 2018). To show that the proposed method also benefits other GNNs/tasks, we present additional experiment results in Appendix B.4 on more GNNs with link prediction task. The experimental results have confirmed the superb performance of the proposed hashing-based coding scheme comparing to the random coding scheme used in `ALONE` under the intended use scenario.

## 2 RELATED WORK

The embedding compression problem is extensively studied for NLP models because of the memory cost associated with storing the embedding vectors, and one of the most popular strategy is parameter sharing (Suzuki & Nagata, 2016; Shu & Nakayama, 2017; Svenstrup et al., 2017; Takase & Kobayashi, 2020). For example, Suzuki & Nagata (2016) trains a small set of sub-vectors shared by all words called "reference vectors" where each word embedding is constructed by concatenating different sub-vectors together. Both the shared sub-vectors and sub-vector assignments are optimized during the training time. The resulting compressed representation is capable of representing each word compactly, but the training process is memory costly as it still needs to train the full embedding matrix to solve the sub-vector assignment problem. Therefore, the method proposed by Suzuki & Nagata (2016) is not suitable for large-scale data.

Shu & Nakayama (2017) trains a encoder-decoder model (i.e., autoencoder) where the encoder converts a pre-trained embedding into the corresponding compositional code representation and the decoder reconstructs the pre-trained embedding from the compositional code. Once the encoder and decoder are trained, all the pre-trained embeddings are converted to the compact compositional code representation using the encoder; then the decoder can be trained together with the downstream models. Because the memory cost associated with the compositional code is much smaller than the raw embedding and the decoder is shared by all words, the method reduces the overall memory consumption associated with representing words. However, since it also requires training the embeddings prior to training the encoder-decoder, the training process still has high memory cost associated with the conventional embedding training similar to the previous work. In other words, the method proposed by Shu & Nakayama (2017) is also not applicable to extreme scale data.

Svenstrup et al. (2017) represents each word compactly with a unique integer and $k$ floating-point values where $k$ is much smaller than the dimension of the embedding. To obtain the embedding from the compact representation of a word, $k$ hash functions[4] are used to hash the word's associated unique integer to an integer in $[0, c)$ where $c$ is much smaller than the number of words. Next, $k$ vectors are extracted from a set of $c$ learnable "component vectors" based on the output of hash function. The final embedding of the word is generated by computing the weighted sum of the $k$ vectors where the weights are based on the $k$ learnable floating-point values associated with the word. Similar to our work, Svenstrup et al. (2017) also uses hash functions in their proposed method. But, the role of the hash function is different: Svenstrup et al. (2017) uses hash functions for reducing cardinality while we use hash functions to perform LSH. On top of that, as the $k$ learnable floating-point values are associated with each word in the method proposed by Svenstrup et al. (2017), their method has its parameter size grown linearly with respect to the vocabulary size which makes the method not ideal for our application.

The `ALONE` method proposed by Takase & Kobayashi (2020) represents each word with a randomly generated compositional code, and the embedding is obtained by inputting the compositional

---

[4]The hash function used by Svenstrup et al. (2017) is the hash function proposed by Carter & Wegman (1979) for hashing integers.

code to a decoder where the number of learnable parameters in the decoder model is independent of the vocabulary size. The ALONE method checks all the requirements for our application; however, the performance suffers when the vocabulary size increases comparing to autoencoder-based approach (Shu & Nakayama, 2017) as demonstrated in Figure 1 (labeled as "learn"). In contrast, our proposed method has similar performance comparing to autoencoder-based approach (Shu & Nakayama, 2017) but does not require the additional training phases required by the method proposed by Shu & Nakayama (2017).

## 3 METHOD

The proposed method consists of two stages: 1) an encoding stage where each node's compositional code is generated with a hashing-based method and 2) a decoding stage where the decoder is trained in an end-to-end fashion together with the downstream model. Figure 2 shows an example forward pass of the embedding construction process. The binary code is a node's compositional code generated by the hashing-based method (Section 3.1). After the binary code is converted to integer code, the decoder model, which mostly includes $m$ codebooks and a multilayer perceptron (MLP) as described in Section 3.2, generates the corresponding embedding. The memory cost of storing both the compositional codes and the decoder is drastically lower than conventional embedding layer as we shown in Section 5.2.

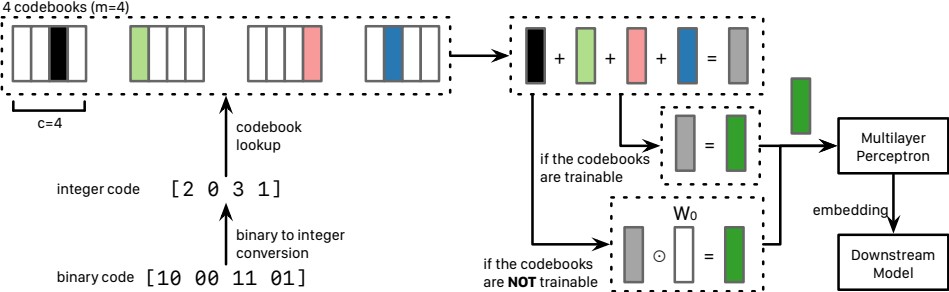

Figure 2: In this toy example, each codebook has 4 vectors ($c = 4$) and there are 4 *distinct* codebooks ($m = 4$). There are two variants of the adopted decoder models: 1) a light version where the codebooks are **NOT** trainable and 2) a full version where the codebooks are trainable. $W_0$ is a trainable vector for rescaling the intermediate representation (see Section 3.2).

### 3.1 HASHING-BASED CODING SCHEME

Algorithm 1 outlines the random projection-based hashing method. The first input to our algorithm is a matrix $\mathbf{A} \in \mathbb{R}^{n \times d}$ containing the auxiliary information of each node where $n$ is the number of nodes and $d$ is the length of auxiliary vector associated with each node. When the adjacency matrix is used as the auxiliary information, $d$ is equal to $n$, and it is preferred to store $\mathbf{A}$ as a sparse matrix in compressed row storage (CRS) format as all the operations on $\mathbf{A}$ are row-wise operations. The other inputs include code cardinality $c$ and code length $m$. These two inputs dictate the format (and the memory cost) of the output compositional code $\hat{\mathbf{X}}$. For each node's associated code vector, $c$ controls the cardinality of each element in the code vector and $m$ controls the length of the code vector. The output is the resulting compositional codes $\hat{\mathbf{X}} \in \mathbb{B}^{n \times m \log_2 c}$ in binary format where each row contains a node's associated code vector where $m \log_2 c$ is the number of bits required to store one code vector. We store $\hat{\mathbf{X}}$ in binary format because binary format is more space efficient comparing to integer format. The binary code vector can be reversed back to integer format before inputting it to the decoder.

In line 2, the number of bits requires to store each code vector (i.e., $m \log_2 c$) is computed and stored in variable $n_{\text{bit}}$. In line 3, a Boolean matrix $\hat{\mathbf{X}}$ of size $n \times n_{\text{bit}}$ is initialized for storing the resulting compositional codes. The default value for the matrix is False. From line 4 to 11, the compositional codes are generated bit-by-bit in the outer loop and node-by-node in the inner loops. Generating compositional codes in such order is a more memory efficient way to perform

---

**Algorithm 1** Encode with Random Projection

**Input:** auxiliary information $\mathbf{A} \in \mathbb{R}^{n \times d}$, code cardinality $c$, code length $m$
**Output:** compositional code $\hat{\mathbf{X}} \in \mathbb{B}^{n \times m \log_2 c}$
1 **function** ENCODE($A, c, m$)
2     $n_{\text{bit}} \leftarrow m \log_2 c$
3     $\hat{\mathbf{X}} \leftarrow$ GETALLFALSEBOOLEANMATRIX($n, n_{\text{bit}}$)
4     **for** $i$ **in** $[0, n_{\text{bit}})$ **do**
5         $V \leftarrow$ GETRANDOMVECTOR($d$)
6         $U \leftarrow$ GETEMPTYVECTOR($n$)
7         **for** $j$ **in** $[0, n)$ **do**
8             $U[j] \leftarrow$ DOTPRODUCT($\mathbf{A}[j, :], V$)
9         $t \leftarrow$ GETMEDIAN($U$)
10         **for** $j$ **in** $[0, n)$ **do**
11             **if** $U[j] > t$ **then** $\hat{\mathbf{X}}[j, i] \leftarrow$ `True`
12     **return** $\hat{\mathbf{X}}$

---

random projection as it only needs to keep a size $d$ random vector in each iteration comparing to the alternative order. If the inner loop (i.e., line 7 to 8) is switched with the outer loop (i.e., line 4 to 11), it would require us to use a $\mathbb{R}^{n_{\text{bit}} \times d}$ matrix to store all the random vectors for random projection (i.e., matrix multiplication).

In line 5, a random vector $V \in \mathbb{R}^d$ is generated; the vector $V$ is used for performing random projection. In line 6, an empty vector $U \in \mathbb{R}^n$ is initialized for storing the result of random projection. From line 7 to 8, each node's associated auxiliary vector is projected using the random vector $V$ and stored in $U$ (i.e., $U = \mathbf{A}V$). Here, the memory footprint could be further reduced if we only load a few rows of $\mathbf{A}$ during the loop instead of the entire $\mathbf{A}$ before the loop. Such optimization could be important as the size of $\mathbf{A}$ could be too large for systems with limited memory. In line 9, the median of $U$ is identified and stored in $t$. This is the threshold for binarizing real values in $U$. From line 10 to 11, using both the values in vector $U$ and the threshold $t$, the binary code is generated for each node. Lastly, in line 12, the resulting compositional codes $\hat{\mathbf{X}}$ are returned. The resulting $\hat{\mathbf{X}}$ can be used for any downstream tasks.

Note, we use the median as the threshold instead of the more commonly seen zero because it reduces the number of collisions in the resulting binary code[5]. Reducing the number of collisions is important for our case because our goal is to generate unique code vector to represent each node. To confirm whether using median as the threshold reduces the number of collisions, we have performed an experiment using pre-trained `metapath2vec` node embeddings (Dong et al., 2017a). We generate the compositional codes with random projection-based hashing with either the median or zero as the threshold. Then, we count the number of collisions in the generated compositional codes. We repeat the experiment for 100 times under two different experimental settings (i.e., 24 bits/32 bits). The experiment results are summarized in Figure 3 with histogram, setting the threshold to median instead of zero indeed reduces the number of collisions. We also repeat the experiments with `metapath2vec++` and `GloVe` embeddings and the conclusion remains the same (see Appendix B.1).

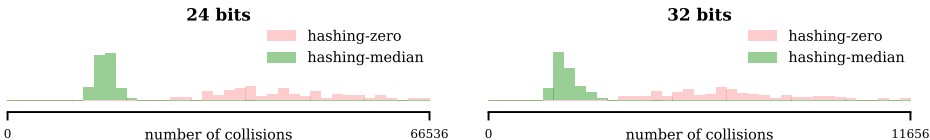

Figure 3: The experiments are performed on `metapath2vec` for 100 times under two different bit length settings: 24 bits and 32 bits. The distribution of the 100 outcomes (i.e., number of collisions) for each method is shown in the figure. The number of collisions is lower for median threshold comparing to zero threshold.

---

[5]The threshold used in the LSH method proposed by Charikar (2002) is zero.

The memory complexity of Algorithm 1 is $O(\text{MAX}(nm \log_2 c, df, nf))$ where $f$ is the number of bits required to store a floating-point number. The $nm \log_2 c$ term is the memory cost associated with storing $\hat{\mathbf{X}}$, the $df$ term is the memory cost associated with storing $V$, and the $nf$ term is the memory cost associated with storing $U$. Because $f$ is usually less than $m \log_2 c$ (i.e., based on hyper-parameters used in prior works[6]) and $d$ is usually less than or equal to $n$, the typical memory complexity of Algorithm 1 is $O(nm \log_2 c)$. In other words, the memory complexity of Algorithm 1 is the same as the output matrix $\hat{\mathbf{X}}$ which shows how memory efficient Algorithm 1 is. The time complexity of Algorithm 1 is $O(nm \log_2 c)$ for the nested loop[7].

### 3.2 DECODER MODEL DESIGN

We will use the example forward pass presented in Figure 2 to introduce the decoder design. The input to the decoder is the binary compositional codes generated from the hashing-based coding scheme introduced in Section 3.1. The input binary code is first converted to integers for using as indexes for retrieving the corresponding real vector from the codebooks. In our example, the binary vector $[10, 00, 11, 01]$ is converted to integer vector $[2, 0, 3, 1]$. Each codebook is a $\mathbb{R}^{c \times d_c}$ matrix where $c$ is the number of codes in the codebook (i.e., code cardinality) and $d_c$ is the size of each real vector in the codebook. There are $m$ codebooks in total where $m$ is the code length (i.e., length of the code after being converted to integer vector from binary vector). Because the code length is 4 in the example, there are 4 codebooks in Figure 2. Because the code cardinality is 4 (i.e., the number of possible values in the integer code), each codebook has 4 real vectors.

From each codebook, a real number vector is retrieved based on each codebook's corresponding index. In our example, the vector corresponding to index 2 (black) is retrieved from the first codebook, index 0 vector (green) is retrieved from the second codebook, index 3 vector (red) is retrieved from the third codebook, and the index 1 vector (blue) is retrieved from the last codebook. The real vectors (i.e., the codebooks) can either be non-trainable random vectors or trainable vectors. We refer to the former method as the *light* method, and the later method as the *full* method. The former method is lighter as the later method increases the number of trainable parameters by $mcd_c$. The full method is desired if the additional trainable parameters (i.e., memory cost) are allowed by the hardware. Note, despite the full method has higher memory cost, the number of trainable parameters still does not depend on the number of nodes in the graph.

Next, the retrieved real vectors are summed together. The summed vector is handled differently for the light and full method. As the codebooks are not trainable for light method, we compute the element-wise product between the summed vector and a trainable vector $W_0 \in \mathbb{R}^{d_c}$ to rescale each dimension of the summed vector following Takase & Kobayashi (2020). Such transformation is not needed for the full method because the full method can capture this kind of transformation with the trainable parameters in the codebooks. The transformed vector is then fed to a MLP with ReLU between linear layers. The output of the MLP is the embedding corresponding to the input compositional code, and the output embedding is fed to the downstream model.

If the number of neurons for the MLP is set to $d_m$, the number of layers for the MLP is set to $l$, and the dimension of the output embedding is set to $d_e$, the light method has $mcd_c$ non-trainable parameters (which can be stored outside of GPU memory) and $d_c + d_c d_m + (l - 2)d_m^2 + d_m d_e$ trainable parameters. The full method has $mcd_c + d_c d_m + (l - 2)d_m^2 + d_m d_e$ trainable parameters. Here, we assume $l$ is greater than or equal to 2. Note, the number of parameters does not grow with increasing number of nodes for both the light and full methods.

## 4 INTEGRATION WITH GRAPHSAGE MODEL

Figure 4 shows how the proposed method is integrated with GraphSage model (Hamilton et al., 2017). The figure depicts a forward pass during training. First, in step 0, a batch of nodes is sampled. In step 1, for each node in the batch, a number of neighboring nodes (i.e., first neighbors) are

---

[6]If single-precision format is used for floating-point numbers, $f$ is 32 bit, and $m \log_2 c$ is commonly set to a number larger than 32 bit in prior works (Shu & Nakayama, 2017; Takase & Kobayashi, 2020).

[7]The median finding algorithm (Blum et al., 1973) in line 9 is $O(n)$ which is the same as the inner loops (i.e., line 7 to 8 and line 10 to 12.).

sampled. Because the example model shown in the figure has 2 layers, the neighbors of neighbors (i.e., second neighbors) are also sampled in step 2. Next, the binary codes associated with each node's first and second neighbors are retrieved in step 3 and decoded in step 4 using the system described in Section 3.2.

After the embeddings for both the first and second neighbors are retrieved, the second neighbor embeddings of each given first neighbor embedding are aggregated with functions like `mean` or `max` in `Aggregate 1` layer. Let's say $H_i$ contains the embeddings of neighboring nodes for a given node $i$, the aggregate layer computes the output $\hat{h}_i$ with $\text{AGGREGATE}(H_i)$. Next, in `Layer 1`, for each first neighbor node $i$, $\hat{h}_i$ and $x_i$ (i.e., embedding for node $i$) are concatenated and processed with a linear layer plus non-linearity. The process of `Layer 1` can be represented with $\sigma(W \cdot \text{CONCATENATE}(\hat{h}_i, x_i))$ where $W$ is the weight associated with the linear layer and $\sigma(\cdot)$ is non-linearity like `ReLU`. Similar process is repeated in `Aggregate 2` layer and `Layer 2` to generate the final representation of each node in the batch. Note, the concatenation step is omitted in `Layer 2` because the node embedding of each node in the batch is not used in `GraphSage` model (Hamilton et al., 2017). The final prediction is computed by feeding the learned representation to the output (i.e., linear) layer. All the parameters in the model are learned end-to-end using the ground truth labels from the training data.

Figure 4: The proposed method can be integrated with the `GraphSage` model. The *Code Lookup* is used to look up the corresponding binary code for each input node. The *Decoder* is the system presented in Figure 2 and converts the input binary codes to embeddings.

## 5 EXPERIMENT

We perform two sets of experiments: 1) pre-trained embedding reconstruction and 2) training decoder with `GraphSage` model jointly. The first set of experiments reveals the difference between different methods' compressing capability while the second set of experiments provides the performance measurement on the targeted application. All experiments are conducted in `Python` with `PyTorch` (Paszke et al., 2019). The source code can be downloaded from: `https://www.dropbox.com/s/1mixmhgbg4wiwtd/release.zip`.

### 5.1 PRE-TRAINED EMBEDDING RECONSTRUCTION

In this set of experiments, we compare the compression capability of different compression methods by testing the quality of the reconstructed embedding. The tested methods are the random coding (i.e., baseline method proposed by Takase & Kobayashi (2020)), the learning-based coding (i.e., autoencoder similar to the method proposed by Shu & Nakayama (2017)), and the hashing-based coding (i.e., the proposed method). When applying the hashing-based coding method on the graph dataset, we feed either the original pre-trained embedding (i.e., hashing (pre-trained) in Figure 1) or the adjacency matrix from the graph (i.e., hashing (graph) in Figure 1) into Algorithm 1. We vary the number of compressed entities when testing different methods.

**Dataset:** Three sets of pre-trained embeddings are used in these experiments: 1) the 300 dimension `GloVe` word embedding, 2) the 128 dimension `metapath2vec` node embedding, and 3) the 128 dimension `metapath2vec++` node embedding. The `GloVe` embeddings are tested with word analogy and similarity tasks. The performance measurements for word analogy and similarity task are accuracy and Spearman's rho, respectively. The `metapath2vec`/`metapath2vec++` embeddings are tested with node clustering, and the performance measurement is normalized mutual information. Please see Appendix A.2 and Appendix A.3 for more details regarding the datasets.

**Implementation:** We use the full decoding method in this set of experiments. To train the compression method, we use mean squared error between the input embeddings and the reconstructed embeddings as the loss function following Takase & Kobayashi (2020). The loss function is optimized

with `AdamW` (Loshchilov & Hutter, 2017) with the default hyper-parameter settings in `PyTorch` (Paszke et al., 2019). Because we want to vary the numbers of compressed entities when comparing different methods, we need to sample from the available pre-trained embeddings. Similar to Takase & Kobayashi (2020), we sample based on the frequency[8]. Since different experiments use different numbers of compressed entities, we only evaluate with the same top 5k entities based on frequency similar to Takase & Kobayashi (2020), despite there are more than 5k reconstructed embeddings when the number of compressed entities is greater than 5k. In this way, we have the same test data across experiments with different numbers of compressed entities. The detailed hyper-parameter settings are shown in Appendix A.4.

**Result:** The experiment results are summarized in Figure 1. Note, we use "random" to denote the baseline method (i.e., `ALONE`). When the number of compressed entities is low, the reconstructed embeddings from all compression methods perform similar to using the raw embeddings (i.e., the original pre-trained embeddings). As the number of compressed entities increases, the reconstructed embeddings' performance decreases. The decreasing performance is likely caused by the fact that the decoder model's size does not grow with the number of compressed entities. In other words, the compression ratio increases as the number of compressed entities increases (see Table 3 in Appendix B.2). When comparing different compression methods, we can observe that the quality of the reconstructed embeddings from random coding method drops sharply comparing to other methods (i.e., hashing-based coding and learning-based coding). It is surprising that the hashing-based coding method works as well as the learning-based coding method even if the learning-based coding method uses additional parameters to learn the coding function. When we compare both variants of the proposed coding method (i.e., hashing with pre-trained and hashing with graph, i.e., adjacency matrix), the performance are very similar. This shows how the adjacency matrix from the graph is a valid choice for applying the proposed hashing-based coding method. We have also tested other settings of $c$ and $m$ (see Appendix B.2); the conclusion stays the same.

## 5.2 NODE CLASSIFICATION

To examine the difference between the compression methods in the intended setup, we perform node classification where the decoder is trained together with a `GraphSAGE` model (Hamilton et al., 2017). Because we assume there is no node feature or pre-trained embedding available in our experiment setup, the autoencoder-based method proposed by Shu & Nakayama (2017) is not applicable for this set of experiments. We compare the proposed hashing-based coding method (using adjacency matrices and Algorithm 1 to generate the code) with two baseline methods: random coding method and raw embedding method. The raw embedding method explicitly learns the embeddings together with the `GraphSAGE` model. The raw baseline method can be treated as the upper bound in terms of accuracy because the embeddings are not compressed. the performance measurement is classification accuracy.

**Dataset:** The experiments are performed on the ogbn-arxiv, ogbn-mag, and ogbn-products datasets from Open Graph Benchmark (Hu et al., 2020). As we are more interested in modeling attribute-less graphs, we discard the node features included in the datasets. We convert all the directed graphs to undirected graphs by making the adjacency matrix symmetry. Please see Appendix A.5 for more detailed information regarding the datasets.

**Implementation:** For the downstream model, we use the `PyTorch` implementation of the `GraphSAGE` model (Hamilton et al., 2017; Johnson et al., 2018). Because the node classification problem is a multiclass classification problem, we use the cross entropy loss to train both the decoder and the `GraphSAGE` model together. The loss function is optimized with `AdamW` (Loshchilov & Hutter, 2017) with the default hyper-parameter settings in the `GraphSAGE` implementation (Johnson et al., 2018). We perform the experiments using two different aggregators: mean pooling and max pooling. The detailed hyper-parameter settings are shown in Appendix A.6.

**Result:** According to Table 1, the proposed hashing-based coding method outperforms the random coding method (i.e., `ALONE`) in all tested scenarios which agrees with our findings presented in Figure 1. One possible reason for the random coding method's less impressive performance comparing to the result reported on NLP tasks by Takase & Kobayashi (2020) is related to the number

---

[8]For `GloVe`, frequency means the times of a word occurs in the training data. For `metapath2vec` and `metapath2vec++`, frequency means the times of a node occurs in the sampled metapaths.

of entities compressed by the compression method. In NLP models, embeddings typically represent sub-words instead of words (Vaswani et al., 2017). For example, the transformer model for machine translation adopted by Takase & Kobayashi (2020) has 32,000 sub-words, which is much smaller comparing to even the smallest tested graph dataset (i.e., ogbn-arxiv with 168,390 nodes). In other words, the proposed hashing-based coding method is more effective for compressing larger set of entities comparing to the baseline random coding method.

Table 1: The proposed method outperforms the baseline method (more results in Table 6). We use NC to denote the non-compressed or embedding learning method *without* compression, Rand to denote the random coding method (i.e., `ALONE`), and Hash to denote the proposed hashing coding method. The numbers presented in the table are classification accuracy.

| Aggregator | ogbn-arxiv | | | ogbn-mag | | | ogbn-products | | |
|---|---|---|---|---|---|---|---|---|---|
| | NC | Rand | Hash | NC | Rand | Hash | NC | Rand | Hash |
| Mean pool | 0.6228 | 0.6045 | 0.6259 | 0.3192 | 0.2989 | 0.3387 | 0.7486 | 0.6327 | 0.6414 |
| Max pool | 0.5884 | 0.4407 | 0.6034 | 0.3083 | 0.3050 | 0.3283 | 0.7294 | 0.6986 | 0.7156 |

In terms of memory usage, the compression method is capable of achieving a considerablely good compression ratio. For example, since the ogbn-products dataset has 1,871,031 nodes, it requires 456.79 MB to store the raw embeddings in GPU memory. On the contrary, it only takes the proposed method 28.55 MB to store the binary codes in CPU memory, and the corresponding decoder model only costs 9.13 MB of GPU memory. The compression ratio is 43.75 for the the proposed method's less memory efficient setup (i.e., full model) if we only consider GPU memory usage. For the total memory usage, the compression ratio is 11.74 for the same setup. The complete memory cost breakdowns for each dataset/method are shown in Appendix B.3. We also repeat this set of experiments with other `GraphSAGE` aggregation function, other GNN architectures, and link prediction task. These additional results are presented in Table 10 and Table 11 with the experimental procedure described in paragraphs accompanying these two tables.

## 6 CONCLUSION

In this work, we proposed a hashing-based coding scheme which generates compositional codes for compactly representing nodes in graph datasets. The proposed coding scheme outperforms the prior embedding compressing method which uses a random coding scheme in almost all experiments. On top of that, the performance degradation coming from the lossy compression is minimal as demonstrated in our experiments. Because the proposed embedding compressing method drastically reduces the memory cost associated with embedding learning, it is now possible to jointly train unique embeddings for all the nodes with GNN models on industrial scale graph datasets.

**Potential Impact and Future Directions:** Aside from GNNs, the proposed methods can also be combined with other kinds of models on tasks that require learning embeddings for a large set of entities. For example, it is common to have categorical features/variables with high cardinalities in transaction data, and embeddings are usually used to represent these categorical features (Du et al., 2019; Yeh et al., 2020a;b). Therefore, the proposed method is well suited for building deep learning models for transaction data. Determining the most effective auxiliary information for generating the binary codes should be an interesting direction for transaction data-based fintech applications. The click-through rate prediction problem (Deng et al., 2021) is another interesting direction to explore with our proposed lightweight embedding compression method as click-through rate datasets also contain categorical features with high cardinalities. As a result, the proposed embedding compressing method could potentially address the scalability problems associated with high cardinalities categorical features in many real world applied machine learning problems. In addition to applying the method to different models/applications, it is also interesting to explore other types of auxiliary information from graphs (e.g., higher-order adjacency matrices) for the encoding stage because the codes generated from richer auxiliary information may provide more information for better embedding compression.

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

## A  ADDITIONAL EXPERIMENT DETAILS

Here, we provide more details regarding the experiments we presented in the main text.

### A.1  COMPARISON OF DIFFERENT THRESHOLDS FOR HASHING-BASED CODING

We have comparing the difference between different choice of thresholds when binarizing the real values into binary codes in Section 3.1 with an experiment. Here, we will describe the details of the experiment. The experiment dataset consists of the first 200,000 pre-trained `metapath2vec` or `metapath2vec++` node embeddings downloaded from the supplemental web page constructed by Dong et al. (2017a;b). The dimension of the pre-trained embeddings is 128. Because we repeat the experiments 100 times, we generate 100 seeds to make sure both method use the same basis to perform random projection as the only difference between the two tested methods should be the threshold. In each experimental trial, we first use the seed to generated a random matrix $V \in \mathbb{R}^{128 \times n_{bit}}$. Next, we project the embedding matrix (i.e., the 200,000 × 128 embedding matrix) using $V$; then, we binarize the result matrix using either zero or the median of each row. With the binary codes prepared, we count the number of coalitions for each method. Once all 100 trials are done, the result is presented with a histogram as shown in Figure 3.

### A.2  WORD ANALOGY AND SIMILARITY TASKS

The pre-trained `GloVe` word embeddings are downloaded from the web page created by Pennington et al. (2014b;a). The word embeddings are trained using Wikipedia 2014 and Gigaword 5 dataset (total of 6B tokens).

**Word analogy:** We downloaded a list of word analogy pairs from the repository of `word2vec` (Mikolov et al., 2013a;b). The word analogy pairs are categorized into 14 categories, and the 14 categories are shown below. We also provide an example pair for each category.

- capital-common-countries: `Athens:Greece::Bangkok:Thailand`
- capital-world: `Ankara:Turkey::Apia:Samoa`
- currency: `Japan:yen::Canada:dollar`
- city-in-state: `Chicago:Illinois::Houston:Texas`
- family: `stepbrother:stepsister::man:woman`
- gram1-adjective-to-adverb: `amazing:amazingly::cheerful:cheerfully`
- gram2-opposite: `acceptable:unacceptable::possible:impossible`
- gram3-comparative: `fast:faster::strong:stronger`
- gram4-superlative: `great:greatest::lucky:luckiest`
- gram5-present-participle: `code:coding::read:reading`
- gram6-nationality-adjective: `Korea:Korean::Sweden:Swedish`
- gram7-past-tense: `going:went::paying:paid`
- gram8-plural: `banana:bananas::monkey:monkeys`

- gram9-plural-verbs: `play:plays::vanish:vanishes`

The experiment is performed as described by Mikolov et al. (2013a). Given a word embedding matrix $\mathbf{X}$ and a word analogy pair (e.g., `Athens:Greece::Bangkok:Thailand`), we first prepare a query vector $Q$ with $\mathbf{X}$[Greece] $-$ $\mathbf{X}$[Athens] $+$ $\mathbf{X}$[Bangkok]. Next, we use $Q$ to query $\mathbf{X}$ with cosine similarity. The answer is only considered correct if the most similar word is `Thailand`. The performance is measured in accuracy. We compute the accuracy for each category; then, we report the average of the 14 accuracy values as the performance for word analogy.

**Word similarity:** Thirteen word similarity datasets are downloaded from the repository of `MUSE` (Conneau et al., 2017a;b; Lample et al., 2017). Each dataset consists of a list of paired words and their ground truth similarity scores. We list the 13 datasets below with two example pairs from each dataset.

- MC-30: `car-automobile:3.92, shore-woodland:0.63`
- MEN-TR-3k: `river-water:49.00, carrot-design:2.00`
- MTurk-287: `genius-intellect:4.09, session-surprises:1.81`
- MTurk-771: `agreement-contract:4.48, baby-computer:1.24`
- RG-65: `gem-jewel:3.94, fruit-furnace:0.05`
- RW-STANFORD: `hyperlink-link:9.12, radiators-beginning:0.00`
- SEMEVAL17: `sculpture-statue:3.83, airport-piece:0.08`
- SIMLEX-999: `simple-easy:9.40, new-ancient:0.23`
- VERB-143: `makes-produced:0.72, causes-used:0.13`
- WS-353-ALL: `telephone-communication:7.50, line-insurance:2.69`
- WS-353-REL: `computer-keyboard:7.62, professor-cucumber:0.31`
- WS-353-SIM: `tiger-cat:7.35, listing-proximity:2.56`
- YP-130: `end-terminate:4.00, imitate-highlight:0.167`

The experiment is performed as described by Faruqui & Dyer (2014). First, the cosine similarity between word embeddings for each pair of words in a dataset is computed. Then, the order based on the cosine similarity is compared with the order based on the ground truth similarity scores. The comparison of orders are measured with Spearman's rho. The result Spearman's rhos from the 13 datasets are averaged and reported.

### A.3 NODE CLUSTERING TASK

The pre-trained `metapath2vec` embeddings, the pre-trained `metapath2vec++` embeddings, the association between nodes (i.e., researchers), and the cluster labels (i.e., research area) are downloaded from the web page created by Dong et al. (2017a;b). The node embeddings are trained with AMiner dataset (Tang et al., 2008). There are total of 246,678 labeled researchers from the downloaded dataset. Each researcher is assigned with one of the 8 research areas listed below.

- Computing Systems
- Theoretical Computer Science
- Computer Networks and Wireless Communication
- Computer Graphics
- Human Computer Interaction
- Computational Linguistics
- Computer Vision and Pattern Recognition
- Databases and Information Systems

We use $k$-means clustering algorithm (Lloyd, 1982) to cluster the embedding associated with each researcher; then, we measure the clustering performance with normalized mutual information.

### A.4 HYPER-PARAMETER SETTING FOR SECTION 5.1

We use the following hyper-parameter settings for the decoders. For `GloVe`, we use $l = 3$, $d_c = d_m = 512$, $d_e = 300$, $c = 2$, and $m = 128$. For `metapath2vec/metapath2vec++`, we use $l = 3$, $d_c = d_m = 512$, $d_e = 128$, $c = 2$, and $m = 128$. Note, the decoder design is the same across different coding schemes tested on the same dataset. We use different $d_e$ for different embeddings because the dimensionality of different pre-trained embeddings is different. The default

hyper-parameter settings for `AdamW` (Loshchilov & Hutter, 2017) in `PyTorch` (Paszke et al., 2019) are: learning rate $= 0.001$, $\beta_1 = 0.9$, $\beta_2 = 0.999$, and weight decay $= 0.01$. We train all models for 1,024 epochs with batch size of 512.

## A.5 NODE CLASSIFICATION TASK

In this section, we provide more details regarding how we process the graph datasets used in Section 5.2. Given a graph dataset, we first discarded the node features come with the dataset. Next, for all the directed graphs, we convert them to undirected graphs. Although ogbn-mag dataset is a heterogeneous graph, we only use the citing relation between paper nodes as the labels are associated with paper nodes. Because we only use the structure of the graph (i.e., node features are discarded), we removed validation/test nodes that is not also presented in training data. In other words, the node classification experiment is performed in semi-supervised learning settings where the validation/test nodes are presented in the training data without their associated labels. After aforementioned process, the statistics associated with the datasets are presented in Table 2.

Table 2: The number of nodes in the training/validation/test set of each dataset.

|  | training | validate | test | total |
|---|---|---|---|---|
| obgn-arxiv | 90941 | 29431 | 48018 | 168390 |
| obgn-mag | 629571 | 64481 | 41706 | 735758 |
| obgn-products | 195639 | 38483 | 1636909 | 1871031 |

## A.6 HYPER-PARAMETER SETTING FOR SECTION 5.2

We use the following hyper-parameter settings for the decoders: $l = 3$, $d_c = d_m = 512$, $d_e = 64$, $c = 256$, and $m = 16$. We evaluated both light and full methods for the decoder, and reported the evaluation accuracy from the method with the best validation accuracy. We use the following hyper-parameter setting for the `GraphSAGE` model: number of layers $= 2$, number of neurons $= 128$, activation function $=$ `ReLU`, and number of neighbors $= 15$. We use the following hyper-parameter settings for the `AdamW` optimizer (Loshchilov & Hutter, 2017): learning rate $= 0.01$, $\beta_1 = 0.9$, $\beta_2 = 0.999$, and weight decay $= 0$. These settings are the default hyper-parameter settings from either the `GraphSAGE` implementation (Johnson et al., 2018) or `PyTorch` (Paszke et al., 2019). We train all models for 10 epochs with batch size of 256, and report the evaluation accuracy from the epoch with the best validation accuracy.

# B ADDITIONAL EXPERIMENT RESULTS

Here, we provide additional experimental results which supplement Section 5.

## B.1 COMPARISON OF DIFFERENT THRESHOLDS FOR HASHING-BASED CODING

When we introduce our hashing-based coding method in Section 3.1, we talk about how the choice of threshold could affect the number of collisions. Specifically, we compare our choice (which uses the median as the threshold) with the more commonly seen threshold of zero. We demonstrate that the median threshold has less collisions comparing to the zero threshold with the experiment presented in Figure 3. Here, we perform additional experiments use `metapath2vec` node embeddings (Dong et al., 2017a). The results for `metapath2vec++` are summarized in Figure 5, and the conclusion is the same: using the median as the threshold for binarization has lower number of collisions comparing to the zero threshold.

We also perform experiments on `GloVe` word embeddings. We use the 300 dimension variant downloaded from the web page created by Pennington et al. (2014b;a). We perform the experiment under four different bit settings: 20 bits, 24 bits, 28 bits, and 32 bits, and the conclusion agrees with our finding presented in Figure 3 and Figure 5.

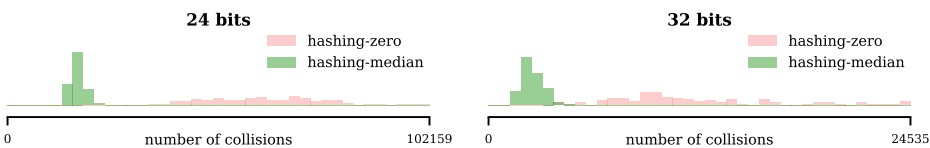

Figure 5: The experiment is performed on `metapath2vec++` for 100 times under two different bit length settings: 24 bits and 32 bits. The distribution of the 100 outcomes (i.e., number of collisions) for each method is shown in the figure. The number of collisions is lower for median threshold comparing to zero threshold.

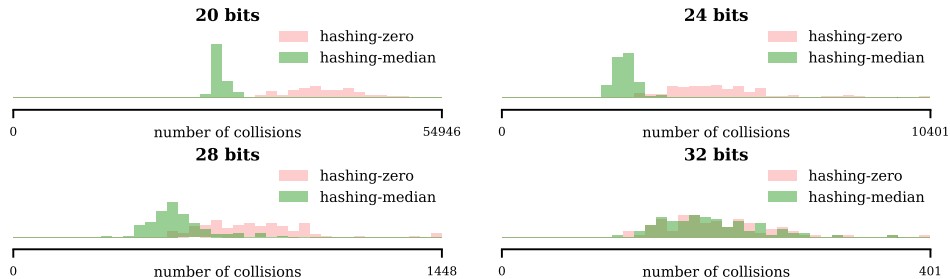

Figure 6: The experiment is performed for 100 times in four different scenarios with `GloVe` embedding: 1) 20 bit code, 2) 24 bit code, 3) 28 bit code, 4) 32 bit code. The distribution of the 100 outcomes (i.e., number of collisions) for each method is shown in the figure. The number of collisions is lower for median threshold comparing to zero threshold.

### B.2 PRE-TRAINED EMBEDDING RECONSTRUCTION

To understand the relationship between the number of compressed entities and the compression ratio, we construct Table 3 to demonstrate how the compression ratio changes as the number of compressed entities is increased.

Table 3: Compression ratios for different numbers of compressed entities. The compression ratios of `metapath2vec++` are omitted as the compression ratios are the same as `metapath2vec`.

| # of Entities | 5000 | 10000 | 25000 | 50000 | 100000 | 200000 |
|---|---|---|---|---|---|---|
| GloVe | 2.65 | 5.11 | 11.60 | 20.09 | 31.69 | 44.55 |
| metapath2vec | 1.34 | 2.57 | 5.73 | 9.72 | 14.91 | 20.34 |

Aside from the results presented in Figure 1, we perform additional experiments to compare the proposed hashing-based coding method with the baseline random coding method under different settings of $c$ and $m$ while varying the number of compressed entities. The results are presented in Table 4. The proposed hashing-based coding method almost always performs better than the baseline random coding method. The performance gap between the two methods increases as the number of entity compressed by the compression method increases. Because the settings of $c$ and $m$ also controls the size of the decoder model, $c$ and $m$ affect the compression ratio. Table 5 shows the compression ratio under different settings of $c$ and $m$. Generally, settings with a lower compression ratio have better performance as the potential information loss is less. In the experiments, the bit size of the binary code is fixed to 128 bits. In other words, both $\{c = 256, m = 16\}$ and $\{c = 2, m = 128\}$ uses 128 bit binary codes. The $c$ and $m$ change the compression ratio by changing the decoder size. When using the $\{c = 256, m = 16\}$ setting, there will be 4,096 vectors total stored in 16 codebooks. When using the $\{c = 2, m = 128\}$ setting, there will be 256 vectors total stored in 2 codebooks. Because the $\{c = 256, m = 16\}$ setting has a larger model (i.e., lower compression ratio), it usually is the setting that outperformed the other in terms of embedding quality. To select a suitable setting for $\{c, m\}$, we suggest the user compute the potential memory

usage and compression ratio for different settings of $\{c, m\}$, then select the one with the lowest compression ratio while still meets the memory requirement.

Table 4: Experiment results on pre-trained embeddings with different settings of $c$ and $m$. We use *random* to denote the random coding method (i.e., `ALONE`), and *hashing* to denote the proposed hashing coding method.

| Embedding/Task | $c$ | $m$ | Coding Method | # of Entities | | | |
|---|---|---|---|---|---|---|---|
| | | | | 5000 | 10000 | 50000 | 200000 |
| GloVe (analogy) | 2 | 128 | random | 0.5783 | 0.4443 | 0.0736 | 0.0048 |
| | | | hashing | 0.5798 | 0.4898 | 0.3638 | 0.2878 |
| | 4 | 64 | random | 0.5933 | 0.4603 | 0.0952 | 0.0066 |
| | | | hashing | 0.6012 | 0.4873 | 0.3204 | 0.2943 |
| | 16 | 32 | random | 0.6209 | 0.5359 | 0.1508 | 0.0134 |
| | | | hashing | 0.6247 | 0.5001 | 0.3596 | 0.2596 |
| | 256 | 16 | random | 0.6708 | 0.6534 | 0.4263 | 0.0837 |
| | | | hashing | 0.6683 | 0.6693 | 0.4710 | 0.3138 |
| GloVe (similarity) | 2 | 128 | random | 0.5435 | 0.5172 | 0.3711 | 0.1056 |
| | | | hashing | 0.5442 | 0.5392 | 0.5260 | 0.4110 |
| | 4 | 64 | random | 0.5799 | 0.5484 | 0.4298 | 0.2218 |
| | | | hashing | 0.5795 | 0.5232 | 0.4837 | 0.4098 |
| | 16 | 32 | random | 0.5495 | 0.5814 | 0.4502 | 0.1620 |
| | | | hashing | 0.5503 | 0.5302 | 0.4465 | 0.4301 |
| | 256 | 16 | random | 0.5743 | 0.5671 | 0.5246 | 0.3613 |
| | | | hashing | 0.5744 | 0.5744 | 0.5309 | 0.4346 |
| metapath2vec | 2 | 128 | random | 0.7730 | 0.7636 | 0.7228 | 0.6026 |
| | | | hashing (pre-trained) | 0.7731 | 0.7650 | 0.7562 | 0.7419 |
| | | | hashing (graph) | 0.7787 | 0.7676 | 0.7468 | 0.7167 |
| | 4 | 64 | random | 0.7720 | 0.7690 | 0.7271 | 0.6273 |
| | | | hashing (pre-trained) | 0.7804 | 0.7697 | 0.7511 | 0.7509 |
| | | | hashing (graph) | 0.7769 | 0.7719 | 0.7526 | 0.7167 |
| | 16 | 32 | random | 0.7763 | 0.7717 | 0.7366 | 0.6686 |
| | | | hashing (pre-trained) | 0.7757 | 0.7668 | 0.7528 | 0.7403 |
| | | | hashing (graph) | 0.7759 | 0.7786 | 0.7638 | 0.7418 |
| | 256 | 16 | random | 0.7790 | 0.7814 | 0.7623 | 0.7263 |
| | | | hashing (pre-trained) | 0.7793 | 0.7767 | 0.7735 | 0.7575 |
| | | | hashing (graph) | 0.7806 | 0.7796 | 0.7601 | 0.7489 |
| metapath2vec++ | 2 | 128 | random | 0.7549 | 0.7585 | 0.7163 | 0.5799 |
| | | | hashing (pre-trained) | 0.7590 | 0.7567 | 0.7361 | 0.7264 |
| | | | hashing (graph) | 0.7536 | 0.7497 | 0.7335 | 0.7013 |
| | 4 | 64 | random | 0.7620 | 0.7480 | 0.7260 | 0.6126 |
| | | | hashing (pre-trained) | 0.7614 | 0.7464 | 0.7380 | 0.7119 |
| | | | hashing (graph) | 0.7593 | 0.7528 | 0.7396 | 0.7025 |
| | 16 | 32 | random | 0.7549 | 0.7499 | 0.7151 | 0.6441 |
| | | | hashing (pre-trained) | 0.7648 | 0.7516 | 0.7463 | 0.7312 |
| | | | hashing (graph) | 0.7610 | 0.7561 | 0.7424 | 0.7270 |
| | 256 | 16 | random | 0.7628 | 0.7640 | 0.7461 | 0.7055 |
| | | | hashing (pre-trained) | 0.7601 | 0.7664 | 0.7495 | 0.7432 |
| | | | hashing (graph) | 0.7656 | 0.7635 | 0.7471 | 0.7292 |

## B.3 NODE CLASSIFICATION TASK

In this section, we provide additional results for different hyper-parameter settings (i.e., different $c$ and $m$). The results are presented in Table 6. The proposed method outperforms the baseline random coding method in all tested cases. The effect of $c$ and $m$ is similar to the conclusion draw from Table 4, and please refer to Appendix B.2 for suggestion on how to select a suitable setting of $c$ and $m$ for different situations. We also provide the memory cost breakdowns for different datasets. We need to compute the memory breakdown for different datasets because the number of nodes in each dataset is different. The breakdowns are shown in Table 7, Table 8, and Table 9. The unit for memory is megabyte (MB), and the column label "ratio" stands for "compression ratio".

Table 5: Compression ratios for different numbers of compressed entities with different settings of $c$ and $m$. The compression ratios of `metapath2vec++` are omitted as the compression ratios are the same as `metapath2vec`.

| Embedding | $c$ | $m$ | # of Entities | | | |
|---|---|---|---|---|---|---|
| | | | 5000 | 10000 | 50000 | 200000 |
| GloVe | 2 | 128 | 2.65 | 5.11 | 20.09 | 44.55 |
| | 4 | 64 | 2.65 | 5.11 | 20.09 | 44.55 |
| | 16 | 32 | 2.15 | 4.18 | 17.09 | 40.60 |
| | 256 | 16 | 0.59 | 1.18 | 5.53 | 18.11 |
| metapath2vec | 2 | 128 | 1.34 | 2.57 | 9.72 | 20.34 |
| | 4 | 64 | 1.34 | 2.57 | 9.72 | 20.34 |
| | 16 | 32 | 1.05 | 2.03 | 8.10 | 18.42 |
| | 256 | 16 | 0.26 | 0.52 | 2.44 | 7.94 |

Table 6: The proposed hashing-based coding outperforms the baseline random coding under different settings of $c$ and $m$. The non-compressed is the embedding learning method *without* compression. We use *random* to denote the random coding method (i.e., `ALONE`), and *hashing* to denote the proposed hashing coding method.

| Aggregator | $c$ | $m$ | Coding Method | ogbn-arxiv | obgn-mag | obgn-products |
|---|---|---|---|---|---|---|
| Mean pool | - | - | non-compressed | 0.6228 | 0.3192 | 0.7486 |
| | 2 | 128 | random | 0.2200 | 0.1402 | 0.5669 |
| | | | hashing | 0.4177 | 0.3366 | 0.6678 |
| | 4 | 64 | random | 0.2337 | 0.0312 | 0.5694 |
| | | | hashing | 0.4233 | 0.3392 | 0.6739 |
| | 16 | 32 | random | 0.4375 | 0.2519 | 0.6499 |
| | | | hashing | 0.5130 | 0.3292 | 0.6960 |
| | 256 | 16 | random | 0.6045 | 0.2989 | 0.6327 |
| | | | hashing | 0.6259 | 0.3387 | 0.6414 |
| Max pool | - | - | non-compressed | 0.5884 | 0.3083 | 0.7294 |
| | 2 | 128 | random | 0.3802 | 0.2188 | 0.5386 |
| | | | hashing | 0.4609 | 0.3138 | 0.6305 |
| | 4 | 64 | random | 0.3762 | 0.2299 | 0.5594 |
| | | | hashing | 0.4906 | 0.3271 | 0.6381 |
| | 16 | 32 | random | 0.5427 | 0.2544 | 0.6047 |
| | | | hashing | 0.5845 | 0.3290 | 0.6393 |
| | 256 | 16 | random | 0.4407 | 0.3050 | 0.6986 |
| | | | hashing | 0.6034 | 0.3283 | 0.7156 |

In additional to the more standard mean and max aggregator, we also repeat the experiments with `GraphSage` model on a non-standard attention aggregator. For this set of experiments, we use the following hyper-parameter settings for the decoders: $l = 3$, $d_c = d_m = 512$, and $d_e = 64$. We use validation data to tune the settings of $c$ and $m$. We use the following hyper-parameter setting for the `GraphSAGE` model: number of layers $= 2$, number of neurons $= 128$, activation function $=$ `ReLU`, and number of neighbors $= 15$. We use the following hyper-parameter settings for the `AdamW` optimizer (Loshchilov & Hutter, 2017): learning rate $= 0.01$, $\beta_1 = 0.9$,

Table 7: The memory cost (MB) for models on ogbn-arxiv dataset.

| Method | CPU | | | GPU | | | | CPU+GPU | |
|---|---|---|---|---|---|---|---|---|---|
| | Binary Code | Decoder | Total | Decoder or Embedding | GNN | Total | Ratio | Total | Ratio |
| Raw | 0.00 | 0.00 | 0.00 | 41.11 | 1.34 | 42.45 | 1.00 | 42.45 | 1.00 |
| Hash-Light | 2.57 | 8.00 | 10.57 | 1.13 | 1.34 | 2.46 | 17.22 | 13.03 | 3.26 |
| Hash-Heavy | 2.57 | 0.00 | 2.57 | 9.13 | 1.34 | 10.46 | 4.06 | 13.03 | 3.26 |

Table 8: The memory cost (MB) for models on ogbn-mag dataset.

| Method | CPU | | | GPU | | | | CPU+GPU | |
|---|---|---|---|---|---|---|---|---|---|
| | Binary Code | Decoder | Total | Decoder or Embedding | GNN | Total | Ratio | Total | Ratio |
| Raw | 0.00 | 0.00 | 0.00 | 179.63 | 1.64 | 181.27 | 1.00 | 181.27 | 1.00 |
| Hash-Light | 11.23 | 8.00 | 19.23 | 1.13 | 1.64 | 2.77 | 65.49 | 21.99 | 8.24 |
| Hash-Heavy | 11.23 | 0.00 | 11.23 | 9.13 | 1.64 | 10.77 | 16.83 | 21.99 | 8.24 |

Table 9: The memory cost (MB) for models on ogbn-products dataset.

| Method | CPU | | | GPU | | | | CPU+GPU | |
|---|---|---|---|---|---|---|---|---|---|
| | Binary Code | Decoder | Total | Decoder or Embedding | GNN | Total | Ratio | Total | Ratio |
| Raw | 0.00 | 0.00 | 0.00 | 456.79 | 1.35 | 458.14 | 1.00 | 458.14 | 1.00 |
| Hash-Light | 28.55 | 8.00 | 36.55 | 1.13 | 1.35 | 2.47 | 185.34 | 39.02 | 11.74 |
| Hash-Heavy | 28.55 | 0.00 | 28.55 | 9.13 | 1.35 | 10.47 | 43.75 | 39.02 | 11.74 |

$\beta_2 = 0.999$, and weight decay $= 0$. We train all models for 30 epochs with batch size of 256, and report the evaluation accuracy from the epoch with the best validation accuracy. The experiment result is presented in Table 10. The proposed compression method outperforms the `ALONE` method.

### B.4 MORE GNNS AND LINK PREDICTION TASK

We also compare the proposed method with `ALONE` with other GNNs like `Graph Convolutional Network` (i.e., `GCN`) (Kipf & Welling, 2016), `Simplifying Graph Convolutional Network` (i.e., `SGC`) (Wu et al., 2019), and `Graph Isomorphism Network` (i.e., `GIN`) (Xu et al., 2018). Although the learning process of these GNNs cannot benefit from the reduced memory provided by both `ALONE` and the proposed method[9], this set of experiment still provides an additional test bed for comparing the two methods similar to the experiment presented in Section 5.1. We use the following hyper-parameter settings for the decoders: $l = 3$, $d_c = d_m = 512$, and $d_e = 64$. We use validation data to tune the settings of $c$ and $m$. For `GCN`, we use a two layered structure with hidden dimension of 128, self-loop, and skip connection. For `SGC` and `GIN`, we also use a two layered structure with hidden dimension of 128 with the other hyper parameter set to the default values in the PyG library (Fey & Lenssen, 2019). We use the following hyper-parameter settings for the `AdamW` optimizer (Loshchilov & Hutter, 2017): learning rate $= 0.001$, $\beta_1 = 0.9$, $\beta_2 = 0.999$, and weight decay $= 0.00001$. We train the network for 512 epochs.

In additional to the node classification datasets, we also perform experiments on link prediction datasets (i.e., ogbl-collab and ogbl-ddi) from Open Graph Benchmark (Hu et al., 2020). The experiment result is summarized in Table 11 and it shows the proposed method almost always outperforming the `ALONE` method.

---

[9]`GCN` cannot benefit from both `ALONE` and the proposed method because all embeddings need to be decoded during the learning process at all time unlike the `GraphSAGE` model which only requires decoding embeddings associated with nodes in the current batch.

Table 10: The proposed hashing-based coding outperforms the baseline random coding with `GraphSage` using non-standard attention aggregator. The non-compressed is the embedding learning method *without* compression. We use *random* to denote the random coding method (i.e., `ALONE`), and *hashing* to denote the proposed hashing coding method.

| Coding Method | ogbn-arxiv | obgn-mag | obgn-products |
|---|---|---|---|
| raw | 0.6063 | 0.3097 | 0.7918 |
| random | 0.5008 | 0.2691 | 0.5981 |
| hashing | 0.5662 | 0.3197 | 0.6758 |

Table 11: The proposed hashing-based coding almost always outperforms the baseline random coding with `GNNs` for both node classification and link prediction. The non-compressed is the embedding learning method *without* compression. We use *random* to denote the random coding method (i.e., `ALONE`), and *hashing* to denote the proposed hashing coding method.

| task | dataset | performance measure | `GraphSage` (mean) NC | Rand | Hash |
|---|---|---|---|---|---|
| node classification | ogbn-arxiv | accuracy | 0.6228 | 0.6045 | **0.6259** |
| | ogbn-mag | accuracy | 0.3192 | 0.2989 | **0.3387** |
| | ogbn-products | accuracy | 0.7486 | 0.6327 | **0.6414** |
| link prediction | ogbl-collab | hits@50 | 0.2740 | **0.1966** | 0.1956 |
| | ogbl-ddi | hits@20 | 0.3277 | 0.3043 | **0.3429** |

| task | dataset | performance measure | GCN NC | Rand | Hash |
|---|---|---|---|---|---|
| node classification | ogbn-arxiv | accuracy | 0.5251 | 0.4957 | **0.5437** |
| | ogbn-mag | accuracy | 0.1815 | 0.1146 | **0.3466** |
| | ogbn-products | accuracy | 0.4719 | 0.3594 | **0.4914** |
| link prediction | ogbl-collab | hits@50 | 0.2316 | 0.1647 | **0.1898** |
| | ogbl-ddi | hits@20 | 0.3697 | **0.3399** | 0.3319 |

| task | dataset | performance measure | SGC NC | Rand | Hash |
|---|---|---|---|---|---|
| node classification | ogbn-arxiv | accuracy | 0.6690 | 0.5491 | **0.5809** |
| | ogbn-mag | accuracy | 0.3523 | 0.1839 | **0.3657** |
| | ogbn-products | accuracy | 0.7686 | 0.3767 | **0.4966** |
| link prediction | ogbl-collab | hits@50 | 0.5589 | 0.4790 | **0.5116** |
| | ogbl-ddi | hits@20 | 0.4841 | 0.5575 | **0.5941** |

| task | dataset | performance measure | GIN NC | Rand | Hash |
|---|---|---|---|---|---|
| node classification | ogbn-arxiv | accuracy | 0.5546 | 0.3736 | **0.5263** |
| | ogbn-mag | accuracy | 0.2728 | 0.2011 | **0.3414** |
| | ogbn-products | accuracy | 0.6423 | 0.4396 | **0.5706** |
| link prediction | ogbl-collab | hits@50 | 0.2614 | 0.2086 | **0.2475** |
| | ogbl-ddi | hits@20 | 0.3216 | 0.3536 | **0.3876** |

