# OpenReview forum: "Embedding Compression with Hashing for Efficient Representation Learning in Graph"
_ICLR.cc/2022/Conference — ICLR 2022 Submitted_

### Official Review · Reviewer_J5SK · 2021-10-31

**Correctness:** 4
**Technical Novelty And Significance:** 2
**Empirical Novelty And Significance:** 2
**Recommendation:** 3
**Confidence:** 4

**Main Review:**

Strength
- The paper is well-written and easy to follow.
- Proposed approach can directly hash the adjacency matrix (no need to have the learned embeddings).

Weakness
- The main technical novelty seems the direct hashing of the adjacency matrix, which is graph specific. When pre-trained embeddings are given, the learning-based approach (Shu & Nakayama, 2017) seems sufficient.
- The evaluation is mainly made on the node classification benchmark. Many node classification benchmarks for GNNs are actually associated with features. In fact, in node classification, I would say that rich features are often available (so that the prediction can be based on the given features), and people often use graph information to regularize the prediction.

Suggestion:
In many link prediction tasks (knowledge graph completion, recommender systems), the node features are often unavailable, and the primary learning signal comes from the edge connectivity information. There, the primary approach is to train shallow embeddings, and the key bottleneck is the storage of embedding for each node (especially for an industrial-scale large graph). I'd be much more impressed if you could demonstrate that your approach works well under the link prediction tasks.

**Summary Of The Paper:**

In this paper, the authors present a general and scalable approach to obtain shallow embeddings of many entities. The authors achieve this by first mapping each entity into a code vector using LSH, and then learning a neural decoder on top of the code vectors to reconstruct the original embeddings/adjacency matrix. The authors then apply their approach to node classification problems, where node features are unavailable (the authors ended up artificially removing node features from existing OGB datasets). The superior performance is demonstrated over the random code vector baseline.

**Summary Of The Review:**

Overall, this paper is well-written, but the technical contribution is not sufficient, and the empirical evaluation should be done on a more practically-relevant task of link prediction.

---

> ### Author Response · Authors · 2021-11-20
> **Thanks for the review**
>
> Thanks for the suggestion. Please see our response below:
>
> ### Response to Weakness and Suggestion
> > 1. The main technical novelty seems the direct hashing of the adjacency matrix, which is graph specific. When pre-trained embeddings are given, the learning-based approach (Shu & Nakayama, 2017) seems sufficient.
>
> In the optimal case (i.e., when the pre-trained embeddings are available), the method proposed by Shu & Nakayama (2017) is sufficient (although it might not be the most cost-efficient).
>
> When our method is compared with the method proposed by Shu & Nakayama (2017) in Section 5.1, our method's performance (in terms of acc., rho, and nmi.) is comparable to Shu & Nakayama's method. However, our method does not require pre-train embedding or the additional training step for encoder training. Our method goes to the end-to-end training step directly after the hashing step.
>
> The section where pre-trained embeddings are used is mainly for comparing different encoding methods. As the experiment includes the method proposed by Shu & Nakayama (2017), pre-trained embeddings are required. The experiment shows how good we could get using cheap hash functions (with adjacency matrix) compared to a more costly method (training an encoder/method by Shu & Nakayama) for compression.
>
> The proposed method is mainly developed for graphs. As suggested by Reviewer u7vW, the model could take different inputs to the hash function for graphs or other applications.
>
> > 2. The evaluation is mainly made on the node classification benchmark. Many node classification benchmarks for GNNs are actually associated with features. In fact, in node classification, I would say that rich features are often available (so that the prediction can be based on the given features), and people often use graph information to regularize the prediction.
> >
> > Suggestion: In many link prediction tasks (knowledge graph completion, recommender systems), the node features are often unavailable, and the primary learning signal comes from the edge connectivity information. There, the primary approach is to train shallow embeddings, and the key bottleneck is the storage of embedding for each node (especially for an industrial-scale large graph). I'd be much more impressed if you could demonstrate that your approach works well under the link prediction tasks.
>
> We agree to the reviewer’s point and we think this is a good suggestion. We have performed additional experiments to test our approach on link prediction tasks. The result is added to the updated paper as the new Table 11. The proposed method also shows its value for the link prediction problem.

---

### Official Review · Reviewer_WEHB · 2021-11-01

**Correctness:** 3
**Technical Novelty And Significance:** 3
**Empirical Novelty And Significance:** Not applicable
**Recommendation:** 6
**Confidence:** 3

**Main Review:**

## Positive aspects:

- The proposed method appears simple and effective. It relies on the well-established technique of locally-sensitive hashing and compositional coding.
- The memory efficiency of the method makes it broadly applicable to large-scale graphs.
- The LSH encoding is constructed from prior knowledge (like vertex connectivity) or pre-trained embeddings, therefore it allows exploiting already available information.
- The method can be trained end-to-end together with all subsequent layers.
- The experiments show that there is only a limited loss in performance with respect to the uncompressed baseline, and substantial improvement with respect to ALONE.

## Negative aspects:

The only negative aspect I see concerns unclear parts that I detail right below.

1. In my opinion, the proposed "full" method (which appears to be the main method proposed by the authors, because it is employed in all the experiments) seems to be more similar to the "learn" baseline by Shu and Nakayama, than to ALONE, both in terms of trainable components and performance. While the improvements over ALONE are evident, a little comparison is provided against the "learn" method. For example, since both methods have learnable codebooks, it is unclear to me what is the additional training phase mentioned at the end of Section 2. Is it associated with the training of the encoder in "learn"? I also would like to see explicit comments on whether there are substantial methodological differences or improvement in efficiency for which one would choose the proposed method rather than "learn" one.
2. I could not find specified whether or not the three methods (ALONE, "learn" and the proposed "full") are set with the same hyperparameters.
3. I think it would be great to also see whether or not there is any computational overhead associated with the three methods, and that the practitioner should be aware of.
4. The second part of the section Implementations about the word sampling requires more clarity. The methods are trained to embed the same subset of words or the words are sampled independently? The test set of 5k words is the same for the three methods?
5. When the auxiliary information matrix is constructed from the adjacency matrix, it seems to me that the test set results in a set of the most connected vertices because words are selected based on their frequency. Secondly, the auxiliary information for not high-frequency words is very sparse. I wonder if it is possible that the observed good performance is biased toward this specific test set. Could you comment on this?


**Summary Of The Paper:**

The paper presents a vertex embedding method with good scalability to large graphs.
The proposed method is based on locally-sensitive hashing and compositional coding: a memory-efficient binary representation of each vertex is constructed from a hashing technique and then, when needed, a decoder creates real vector embeddings of the vertices via compositional coding from a pool of codebooks. The encoding is constructed from prior knowledge and requires no training. Conversely, the decoder is trainable and can be learned end-to-end. The experiments show improved performance. A negative note is that some parts are unclear to me.

**Summary Of The Review:**

The method displays clear advantages over ALONE, and has the potential to be applicable in a wide range of problems, especially, large-scale ones.
Although the methodological contribution is little, the method appears simple and effective.
However, given the mentioned unclear part, I tend to recommend a weak accept. I am confident the authors can clarify the raised doubts and I am happy to increase my score, if appropriate.

---

> ### Author Response · Authors · 2021-11-20
> **Thanks for the review (part 1 of 2)**
>
> Thanks for reviewing our paper. Please see our response below:
>
> ### Response to Negative Aspects
> > 1. In my opinion, the proposed "full" method (which appears to be the main method proposed by the authors, because it is employed in all the experiments) seems to be more similar to the "learn" baseline by Shu and Nakayama, than to ALONE, both in terms of trainable components and performance. While the improvements over ALONE are evident, a little comparison is provided against the "learn" method. For example, since both methods have learnable codebooks, it is unclear to me what is the additional training phase mentioned at the end of Section 2. Is it associated with the training of the encoder in "learn"? I also would like to see explicit comments on whether there are substantial methodological differences or improvement in efficiency for which one would choose the proposed method rather than "learn" one.
>
> The overall training process for the "learn" baseline by Shu and Nakayama has four stages: 1) pre-training the embedding, 2) training the encoder and decoder, 3) encoding each embedding with the encoder, and 4) saving only the codes and the decoder for refining with the downstream model. For both ALONE and the proposed method, the overall training process has two stages: 1) generate the codes for each word/node, and 2) train the decoder with the downstream model. The major difference (i.e., time saving) between "learn" and "ALONE"/proposed is the pre-training step. The cost associated with the pre-training embedding step differs depending on the method used to train the embedding. Shu and Nakayama's method could generate higher quality coding for each word/node but takes more time/resources to reach the end result. In short, Shu and Nakayama's method is less practical when the pre-training step is too expensive.
>
> > 2. I could not find specified whether or not the three methods (ALONE, "learn" and the proposed "full") are set with the same hyperparameters.
>
> Yes, we set up them to use the same decoder design and same hyperparameter. The difference is how the code for each node/word is generated. "ALONE" generate random codes (free but at the cost of accuracy), "learn" train an encoder to generate the code (best accuracy but not time efficient), and the proposed method tries to find a good balance between time and accuracy trade-off. Thanks to the reviewer for pointing it out. We have updated Appendix A.4 to include the following sentence in the updated paper: Note, the decoder design is the same across different coding schemes tested on the same dataset.
>
> > 3. I think it would be great to also see whether or not there is any computational overhead associated with the three methods, and that the practitioner should be aware of.
>
> The proposed method has two stages: 1) encoding and 2) decoding stage. The time complexity for the encoding stage is analyzed in Section 3.1, and the complexity is $O(n m \log_2 c)$ where m and c are hyper-parameters (i.e., code length and code cardinality), n is the number of nodes in the graph. In other words, the first stage is linear with respect to the number of nodes in the graph. The major overhead for the second stage comes from the decoder model (i.e., the time complexity is similar to an MLP). The overhead depends on the design of the decoder. But, since the decoder is usually much lighter than the GNN, the overhead on the runtime is not significant.
>
> The major difference between the three methods is in the first stage as all three methods could in theory have the same decoder design.
>
> In comparison with the method proposed by Shu and Nakayama, the proposed method skips the pre-training step. The pre-training step is usually very expensive (i.e., going through the dataset multiple times while updating the embedding). As a result, the overhead of both ALONE and the proposed method is much smaller than the method proposed by Shu and Nakayama.
>
> For ALONE, the encoding step is also linear to the number of nodes. It generates random numbers for each node, which can be slightly faster. On the whole, ALONE and the proposed method are within the same order of magnitude (linear) while the method proposed by Shu and Nakayama requires a costly pre-training step.
>
> see [https://openreview.net/forum?id=ZaI7Rd11G4S&noteId=6nlOpSjcua](https://openreview.net/forum?id=ZaI7Rd11G4S&noteId=6nlOpSjcua) for part 2.

---

> ### Author Response · Authors · 2021-11-20
> **Thanks for the review (part 2 of 2)**
>
> see [https://openreview.net/forum?id=ZaI7Rd11G4S&noteId=r6N4ArTurC0](https://openreview.net/forum?id=ZaI7Rd11G4S&noteId=r6N4ArTurC0) for part 1.
>
> > 4. The second part of the section Implementations about the word sampling requires more clarity. The methods are trained to embed the same subset of words or the words are sampled independently? The test set of 5k words is the same for the three methods?
>
> Across different experiments, the set of tested words/nodes is always the same 5k words/nodes. The set of embedded words/nodes is also the same for different methods. The test word/node set is always a subset of the embedded word set/node.
>
> > 5. When the auxiliary information matrix is constructed from the adjacency matrix, it seems to me that the test set results in a set of the most connected vertices because words are selected based on their frequency. Secondly, the auxiliary information for not high-frequency words is very sparse. I wonder if it is possible that the observed good performance is biased toward this specific test set. Could you comment on this?
>
> Your hypothesis could be true. The more a model knows about a word/node, the more likely that a good representation is learned for the word/node. The main reason for using high frequency words is that word analogy/similarity datasets mostly contain high frequency words.
>
> Note, sampling is only performed in Section 5.1. The sampling is used as a way to control the amount of compressed information (and fix the test set). We do not sample in Section 5.2 and use all nodes in the graph. The train/validation/test split comes with the dataset.

---

> > ### Comment · Reviewer_WEHB · 2021-11-29
> > **Thanks for the clarifications**
> >
> > I thank the authors for their additional clarifications

---

### Official Review · Reviewer_u7vW · 2021-11-01

**Correctness:** 4
**Technical Novelty And Significance:** 3
**Empirical Novelty And Significance:** 4
**Recommendation:** 6
**Confidence:** 3

**Main Review:**

Strength
1. The proposed method significantly reduces the memory cost when training the GNN models by adopting the random projection hashing approach.
2. Experiments on embedding reconstruction and node classification tasks demonstrate the proposed method outperforms the prior embedding compression method based on a random coding scheme.
3. The compressed embedding generated from the proposed method achieves a small loss compared to the raw embedding on the node classification task, which indicates that the proposed method solves the performance degradation problem to a certain extent.

Weakness
1. The first concern is whether the proposed method still works on other graph representation learning models. In this work, the authors only used GraphSAGE with different aggregators in the evaluation. It would be great if the authors could discuss and include more graph representation learning models in the experimental section, such as GIN [1], deeper GNNs (e.g., [2], [3]), and graph representation learning models with greater expressive power than GraphSAGE (e.g., [4], [5]).
2. Second, the experimental section only considers the node classification task, and I wonder if the compressed embedding generated from the proposed method still achieves good performance on other tasks, such as structural role prediction and link prediction?
3. Next, the authors use the adjacency matrix of the input graph as the auxiliary information in the paper. I wonder if other matrices related to graph structure can also be used as the input auxiliary information? For example, is it possible to use a higher-order adjacency matrix (i.e., $A^{k}$) to encode more global graph structure information?
4. Finally, it would be great if the authors could discuss about how to select the code cardinality $c$ and the code length $m$.


[1] Xu, K., Hu, W., Leskovec, J., & Jegelka, S. (2018). How powerful are graph neural networks?. arXiv preprint arXiv:1810.00826.

[2] Xu, K., Li, C., Tian, Y., Sonobe, T., Kawarabayashi, K. I., & Jegelka, S. (2018, July). Representation learning on graphs with jumping knowledge networks. In International Conference on Machine Learning (pp. 5453-5462). PMLR.

[3] Li, G., Muller, M., Thabet, A., & Ghanem, B. (2019). Deepgcns: Can gcns go as deep as cnns?. In Proceedings of the IEEE/CVF International Conference on Computer Vision (pp. 9267-9276).

[4] You, J., Gomes-Selman, J., Ying, R., & Leskovec, J. (2021). Identity-aware graph neural networks. arXiv preprint arXiv:2101.10320.

[5] Li, P., Wang, Y., Wang, H., & Leskovec, J. (2020). Distance encoding: Design provably more powerful neural networks for graph representation learning. arXiv preprint arXiv:2009.00142.

**Summary Of The Paper:**

In this work, the authors propose a hashing-based node embedding compression approach, which utilizes the random projection hashing method to generate a code vector for each node using auxiliary information such as the input graph adjacency matrix. The proposed method is memory-efficient in the training procedures of Graph Neural Networks (GNN) models. Experiments also demonstrate that the proposed method outperforms other coding schemes in both the embedding reconstruction task and node classification task.



**Summary Of The Review:**

In this work, the authors develop a hashing-based node embedding compression method, which significantly reduces the memory cost in the embedding learning procedure. The proposed method also outperforms the prior random coding based embedding compression method in the experiments. However, the experiments in this paper only utilized GraphSAGE as the graph representation learning model and considered the node classification task. It would be great if the authors could discuss and include more graph representation learning models and more graph related tasks in the experimental section. In addition, I wonder if the authors could discuss about the selection of input auxiliary information?

---

> ### Author Response · Authors · 2021-11-20
> **Thanks for the review**
>
> Thanks for the feedback. Please see our response below:
>
> ### Response to Weakness
> > 1. The first concern is whether the proposed method still works on other graph representation learning models. In this work, the authors only used GraphSAGE with different aggregators in the evaluation. It would be great if the authors could discuss and include more graph representation learning models in the experimental section, such as GIN [1], deeper GNNs (e.g., [2], [3]), and graph representation learning models with greater expressive power than GraphSAGE (e.g., [4], [5]).
>
> It is interesting to see how well the proposed method works with models other than GraphSage. We have performed additional experiments with other architectures like GCN [6], SGC [7], and GIN [1]. The results are presented in Table 11 of the updated paper. The proposed method also works with other graph representation learning models.
>
> > 2. Second, the experimental section only considers the node classification task, and I wonder if the compressed embedding generated from the proposed method still achieves good performance on other tasks, such as structural role prediction and link prediction?
>
> We have also added experiments on link prediction datasets. The results are summarized in Table 11 of the updated paper. The proposed method almost always outperforms the baseline. When it loses to the baseline, the difference is relatively small.
>
> > 3. Next, the authors use the adjacency matrix of the input graph as the auxiliary information in the paper. I wonder if other matrices related to graph structure can also be used as the input auxiliary information? For example, is it possible to use a higher-order adjacency matrix (i.e., ) to encode more global graph structure information?
>
> This is an interesting question. We think it could result in better code for datasets that require higher-order information. We have included this as a potential future direction in the conclusion section of the updated paper.
>
> > 4. Finally, it would be great if the authors could discuss about how to select the code cardinality and the code length.
>
> We have added the following passage to the paper on page 15 of the updated paper:
>
> Generally, settings with a lower compression ratio have better performance as the potential information loss is less. In the experiments, the bit size of the binary code is fixed to 128 bits. In other words, both $\{c=256, m=16\}$ and $\{c=2, m=128\}$ uses 128 bit binary codes. The $c$ and $m$ change the compression ratio by changing the decoder size. When using the $\{c=256, m=16\}$ setting, there will be 4,096 vectors total stored in 16 codebooks. When using the $\{c=2, m=128\}$ setting, there will be 256 vectors total stored in 2 codebooks. Because the $\{c=256, m=16\}$ setting has a larger model (i.e., lower compression ratio), it usually is the setting that outperformed the other in terms of embedding quality. To select a suitable setting for $\{c, m\}$, we suggest the user compute the potential memory usage and compression ratio for different settings of $\{c, m\}$, then select the one with the lowest compression ratio while still meets the memory requirement.
>
> ### Reference
> [1] Xu, K., Hu, W., Leskovec, J., & Jegelka, S. (2018). How powerful are graph neural networks?. arXiv preprint arXiv:1810.00826.
>
> [2] Xu, K., Li, C., Tian, Y., Sonobe, T., Kawarabayashi, K. I., & Jegelka, S. (2018, July). Representation learning on graphs with jumping knowledge networks. In International Conference on Machine Learning (pp. 5453-5462). PMLR.
>
> [3] Li, G., Muller, M., Thabet, A., & Ghanem, B. (2019). Deepgcns: Can gcns go as deep as cnns?. In Proceedings of the IEEE/CVF International Conference on Computer Vision (pp. 9267-9276).
>
> [4] You, J., Gomes-Selman, J., Ying, R., & Leskovec, J. (2021). Identity-aware graph neural networks. arXiv preprint arXiv:2101.10320.
>
> [5] Li, P., Wang, Y., Wang, H., & Leskovec, J. (2020). Distance encoding: Design provably more powerful neural networks for graph representation learning. arXiv preprint arXiv:2009.00142.
>
> [6] Kipf, T. N., & Welling, M. (2016). Semi-supervised classification with graph convolutional networks. arXiv preprint arXiv:1609.02907.
>
> [7] Wu, F., Souza, A., Zhang, T., Fifty, C., Yu, T., & Weinberger, K. (2019, May). Simplifying graph convolutional networks. In International conference on machine learning (pp. 6861-6871). PMLR.

---

### Official Review · Reviewer_hQPL · 2021-11-04

**Correctness:** 3
**Technical Novelty And Significance:** 2
**Empirical Novelty And Significance:** 2
**Recommendation:** 3
**Confidence:** 4

**Main Review:**

W1. The proposed method is straightforward, whose technical novelty is kind of limited given the recent works on embedding compression, e.g., [1], [2], [3].

[1] Learning to Hash with Graph Neural Networks for Recommender Systems. Tan et. al., the Web Conf 2020

[2] Differentiable Product Quantization for End-to-End Embedding Compression. Chen et. al. ICML 2020

[3] GHashing: Semantic Graph Hashing for Approximate Similarity Search in Graph Databases. , Qin et. al., KDD 2020

W2. The experiments studies are insufficient, many of the related works are missing from the evaluation, which is hard to support the paper's claimed contribution.

W3. The paper claims that the memory consumption is a bottleneck while training GNNs or graph representation models. Unfortunately, this claim is not true. In fact, the embedding table only needs to be partially loaded to GPU RAM during the training process. We would suggest the author to make reference to [4] for memory efficient implementation of their work.

[4] DGL-KE: training knowledge graph embeddings at scale, Zheng et. al., SIGIR 2020

**Summary Of The Paper:**

This paper studies the embedding compression problem related to GNNs and graph representation. A two-stage method is proposed to generate the compressed embeddings: firstly, it encodes each node into its composite code with hashing; secondly, it uses a MLP module to decode the embedding for the node. Experiments are performed to evaluate the compression effect with both pretrained graph embeddings and node classification task with GraphSage.

**Summary Of The Review:**

This paper works on the embedding compression problem for GNNs and graph representation models. However, the major limitations on technical novelty and experimental studies make it unlike to be a quality publication.

---

> ### Author Response · Authors · 2021-11-20
> **Thanks for the review**
>
> Thanks for the review. Please see our response below:
>
> ### Response to Weakness
> > 1. The proposed method is straightforward, whose technical novelty is kind of limited given the recent works on embedding compression, e.g., [1], [2], [3].
>
> The problems solved by [1] and [3] are different from the problem solved by the proposed method. The method proposed in [1] and [3] are both learning-to-hash methods; its task is mainly for graph similarity search where the binary codes are generated using the sign function [1] or the binary regularization loss [3]. The methods proposed by [1] and [3] focus on training a GNN-based encoder to compress the hidden representation/embedding, but the input features remain unchanged. In our scenario, the goal is to efficiently compress the input feature/embedding without any embedding/encoder pre-training step. Indeed, the proposed method can be seamlessly compatible with [1,3], like compressing input and hidden embeddings at the same time. We leave this extension in the future.
>
> The method presented in [2] requires the embedding table (query matrix, Q) as an input for the training process. In other words, it also does not apply to our problem setting. We are interested in the case where the embeddings are not available before the training stage (i.e., the training stage where the decoder and the downstream model are trained together).
>
> > 2. The experiments studies are insufficient, many of the related works are missing from the evaluation, which is hard to support the paper's claimed contribution.
>
> We have included additional experiments based on the reviewer’s suggestion. Note that our work proposed the embedding compression approach which is followed by a graph neural network. To our knowledge, we have included similar efforts including a pre-training-based approach and a random coding-based approach. We added experiments to validate the effectiveness of our proposed embedding compression method with several different graph neural networks, including GCN, GIN, and SGC. We also added link prediction experiments where node features are often missing.
>
> > 3. The paper claims that the memory consumption is a bottleneck while training GNNs or graph representation models. Unfortunately, this claim is not true. In fact, the embedding table only needs to be partially loaded to GPU RAM during the training process. We would suggest the author to make reference to [4] for memory efficient implementation of their work.
>
> The system outlined in [4] is specifically designed for knowledge graph embeddings models trained with batches of triplets (head, relation, tail). On top of it, the DGL-KE system [4] mainly focuses on the distributed training, not on the embedding compression.
>
> Even so, it is still non-trivial to set up the distributed training schedule. In practice, setting up large-scale end-to-end graph learning is still largely constrained by memory usage, especially in production. Our proposed approach improves upon the existing compression method, and achieves comparable results to non-compression models, while drastically reducing memory usage.
>
> ### Reference
> [1] Learning to Hash with Graph Neural Networks for Recommender Systems. Tan et. al., the Web Conf 2020
>
> [2] Differentiable Product Quantization for End-to-End Embedding Compression. Chen et. al. ICML 2020
>
> [3] GHashing: Semantic Graph Hashing for Approximate Similarity Search in Graph Databases. Qin et. al., KDD 2020
>
> [4] DGL-KE: training knowledge graph embeddings at scale, Zheng et. al., SIGIR 2020

---

### Decision · Program_Chairs · 2022-01-20

**Decision:**

Reject

**Comment:**

This paper studies the embedding compression problem related to GNNs and graph representation. A two-stage method is proposed to generate the compressed embeddings: firstly, it encodes each node into its composite code with hashing; secondly, it uses a MLP module to decode the embedding for the node. Experiments are performed to evaluate the compression effect with both pretrained graph embeddings and node classification tasks with GraphSage.

The paper considers hashing/compressing the rows/columns of adjacency matrices and using the compressed rows of the adjacency matrices as node features. The adjacency matrices are  intrinsically redundant. Therefore, it is unclear whether the achieved compression rate is significant, especially when applied to settings with known node features.

Some reviewers pointed out existing methods on learning-to-hash methods, which train a GNN-based encoder to compress the hidden representation/embedding, are relevant. Although the authors claim that in their scenario, the goal is to efficiently compress the input feature/embedding without any embedding/encoder pre-training step, it is unclear how the proposed method compares with the learning-to-hash methods when considering the adjacency matrices as the auxiliary information.

The dependence on the number of nodes is also a concern in terms of scalability, as we know the bottleneck of scalability in GNNs is the number of nodes.

The authors use the adjacency matrix of the input graph as the auxiliary information in the paper, which only considers local structure information. The reviewers are curious whether this approach would work for tasks in which global graph structure information is required.

On a minor note, the reviewers also think that the paper would be stronger if the authors provide more principled guidance on how to select the code cardinality c and the code length m.